# Effect of the red uniform on the judgment of position or movement used in Wushu Routine, evaluated by practitioners of the modality

Jinkun Li[1,2], Jingmin Zhang[1], Shuo Tao[1], Xiaoying Zeng[3], Rong Zou[1,4], Xiaobin Hong[1,4]*

1 Department of Psychology, Wuhan Sports University, Wuhan, China, 2 School of Physical Education and Sports, Central China Normal University, Wuhan, China, 3 Department of Wushu, Wuhan Sports University, Wuhan, China, 4 Hubei Key Laboratory of Exercise Training and Monitoring, College of Sports Medicine, Wuhan Sports University, Wuhan, China

* 471780693@qq.com

**Data Availability Statement:** All raw data files are available from the "Zenodo" database (accession number: 10325917).

## Abstract

In the artistic sports program, the referee' scores directly determine the final results of the athletes. Wushu is a artistic sport that has a Chinese characteristic and has the potential to become an official competition at the Summer Olympic. In this study we tested whether a red uniform color affects Wushu Routine practitioners' ratings of athletes' position or movement of Wushu Routine. We also tested whether the effect varied depending on the gender of the athlete and the practitioner, and depending on whether female practitioners were in the ovulation phase of their menstrual cycle. Male (Experiment 1: $N = 72$) and female (Experiment 1: $N = 72$; Experiment 2: $N = 52$) participants who major in Wushu Routine were recruited to take a referee's perspective and rate the movement quality of male and female athletes wearing red or blue uniforms. The results of Experiment 1 showed that both male and female athletes wearing red uniform (compared to blue uniform) received higher ratings ($p = .002$, $\eta^2 = .066$; $p = .014$, $\eta^2 = .043$), and the red effect was especially strong when male practitioners rated female athletes ($p = .002$, $\eta^2 = .069$). The results of Experiment 2, in an all-female sample, showed that in most cases there was no difference in ratings made by women in the ovulation and non-ovulation phases of their menstrual cycle, with the exception of their ratings of male athletes wearing red; in this condition, women gave higher ratings when they were in the ovulation phase of their cycle ($p = .026$). The results suggest that there is a red effect in an artistic sport like Wushu Routine, in which gender and the female menstrual cycle play an important role.

## 1. Introduction

Chinese Traditional Wushu has its origins in hunting and warfare and is deeply rooted in Chinese culture. On January 8th, 2020, the International Olympic Committee announced that

**Funding:** This project is financially supported by the 14th Five-Year Plan Advantageous and Characteristic Disciplines (Groups) of Colleges and Universities in Hubei Province for Exercise and Brain Science ((Hubei Teaching Research [2021] No. 5); Major Program of Philosophy and Social Science Research of Hubei Provincial Department of Education (21ZD095). Funder is Hubei Provincial Department of Education, URL is jyt.hubei.gov.cn, grant recipient is Xiaobin Hong (XBH). The funders had no role in study design, data collection and analysis, decision to publish, or preparation of the manuscript.

Wushu would become an official competitive sport in the 2022 Dakar Youth Olympic Games. The Wushu events are divided into two categories: Changquan and Taijiquan, both of which are types of Wushu Routine. Wushu Routine has been systematically recreated from its origins to become a form of sport that integrates technical skill and artistry to better and more poetically express stylized aggression [1]. The primary condition for Wushu Routine to enter the Summer Olympic Games is that competition rules and the scoring system align with the Olympic standards to ensure fairness and impartiality. Competitive Wushu Routine has incorporated rules and scoring methods from modern aesthetic sports, such as gymnastics, artistic gymnastics, and figure skating [2]. It was anticipated that Wushu Routine would gain early entry into the Summer Olympic Games.

Currently, researchers and Wushu Routine practitioners are primarily focused on judges' ratings of the quality of movements, rehearsal level, difficult movements, and competition process. However, relatively little attention has been paid to the athlete's uniform and the colors of that uniform as influences on the scores awarded to the athlete. Competitive Wushu Routine uniform color choice is generally free within the limits of the rules and is matched according to event characteristics and personal preference, with no standardization. Does the color of a Wushu Routine uniform impact the referees' evaluation of athletes? Can evolutionary psychology and social psychology predict whether the effect of color is influenced by the referee's gender, and among female raters, do evaluations vary based on the menstrual cycle?

Over the past nearly two decades, many researchers have investigated the impact of competition uniform color in competitive sports. These studies serve as the foundation for testing color effects in competitive Wushu Routine. Hill & Barton found that athletes wearing red had significantly higher win rates than those wearing blue in four competitive sports, namely boxing, tae kwon do, Greco-Roman wrestling, and freestyle wrestling at the 2004 Athens Olympics [3]. Hill & Barton attributed this phenomenon to the red-aggressiveness association, whereby red is associated with greater dominance and aggression. Their study triggered a wave of research on the color effect of competitive uniforms, and in the following years, a study demonstrated that there was also a uniform color effect in soccer, whereby teams wearing red uniforms had a higher win rate than teams wearing other colors [4]. A study have also analyzed the advantages of a red uniform in China's Wushu sanda, the result showed that athletes wearing red had a 65% win rate [5].

However, these findings were challenged by researchers who analyzed the winning percentage of uniform colors in a range of competitive sports including taekwondo, international wrestling, soccer, rugby, and basketball. The results showed that there was no "red win effect" or that the"red win effect" disappeared after controlling for other factors affecting the game [6–12]. Therefore, there is ongoing debate regarding whether the color of competitive uniforms affects winning. This may be because the color effect could be subtle and overridden by other influences, such as the strength of the two teams and the athletes' physical state, or even external factors such as time and weather.

Some scholars have turned their focus to a more sensitive indicator of the red effect, namely referee judgments. There have been inconsistent results. On one hand, Hagemann et al. demonstrated that taekwondo judges gave higher scores to athletes wearing red protective gear, and the same was observed when the two athletes switched their protective gear [13]. This may be attributed to referees' tendency to perceive athletes wearing red gear as more dominant and aggressive. In recent years, electronic protective gear has gradually been introduced in taekwondo competitions, and the application of this technology has led to more objective scoring by referees. Researchers who analyzed competition data showed that this new scoring method may have weakened the red effect on match outcomes [14].

On the other hand, researchers focused on team sports have reported that red uniforms may have a negative impact on referees' judgments [15]. Although referees play an essential role in these sports, the final decision of who wins or loses is not entirely in their hands. In contrast, aesthetic sports rely entirely on the referees' scores to determine the final outcome. Therefore, it is important to investigate how uniform color affects the referees' scores in aesthetic sports to promote fair competition in such events. The Changquan type of Wushu Routine is an aesthetic sport in which the movements showcase the Chinese expression of rhythmic beauty. The athlete pursues excellence in the external abilities of stretching, flexibility, speed, and strength, and pursues excellence in the internal qualities of essence, qi and spirit. Although the Changquan referees' ratings of the athletes are not meant to be attractiveness ratings, the referees' ratings of the athletes are likely influenced by athletes' attractiveness.

Research to date suggests that part of an athlete's attractiveness may be in part due to the color of their uniform. According to Elliot's color-in-context (CIC) theory, color conveys specific meanings beyond aesthetics based on innate biological predispositions and acquired social learning. The theory holds that colors trigger automatic assessment processes of which the assessor is unaware [16]. A scholar investigated whether the athletes' uniform color affects referees' scores in other. The results of a study on another aesthetic sport, rhythmic gymnastics, support this idea. Both professional and novice referees viewed athletes wearing red uniforms as performing better than athletes wearing other colors; moreover, the referees had different EEG responses when they looked at athletes wearing red and other uniform colors [17].

However, no studies to date have tested whether ratings of attractiveness based on the color of an athlete's uniform vary depending on the gender of the athlete or the gender of the referee. Previous studies have shown a red-attractiveness effect on mutual evaluations between members of the opposite sex. For men, a red border around a woman's photo, red clothing, or red objects carried by a woman can enhance men's evaluation of the woman's attractiveness [18–20], and men behave more intimately towards them [21]. Women also perceive a man as more attractive when their photo is on a red background or they are wearing red clothing [22], and in one study women rushed more to arrive when they learned that their date was wearing red clothing [23]. Men's and women's responses to the color red may be based both on biology and social learning. In nonhuman primates, red coloring of the skin or genitals can signal fertility of females [24,25] and The status of males [26]. For human, although primitive traits are gradually disappearing, while psychology resulting from evolution may still exist. Men came to interpret the red color on women as sexual receptivity [27], and women came to interpret the red color on men as indicative of status, dominance, and health [19,22,28,29]. Men and women are often not conscious of the red effect. Similarly, referees of Wushu Routine may be unaware of their interpretations of the color red on uniforms, and the effects of these interpretations on scoring.

In addition, female referees' physiological peculiarities may affect a women's perception of red. Women are fertile for only a few days during of their menstrual cycle. According to the ovulatory shift hypothesis, during this time, women's mating psychology, preferences, behaviors, and motivations change in ways that enhance their mating outcomes [30,31]. In terms of heterosexual preference, women prefer men with typically masculine features, such as more masculine faces [32,33] and low voice pitch [34] at ovulation; In terms of personal behavior, ovulating women have been shown to engage in more sexual activity [35,36], respond more favorably to a courtship request [37], and use more cosmetics and spend more time getting dressed [38,39]. These behaviors may stem from the reproductive motivation that fluctuates with the menstrual cycle, a pattern that might have evolutionarily adaptive [40,41]. This reproductive motivation drives women to attract and focus on potential high-quality mates to increase the likelihood of selecting superior genetic heritage. In previous research on the red

effect, woman perceive red clothing on men was shown to be associated with health [29], social status [22] and dominance [28], all of which might signal high-quality genes for women choosing a potential mate. Therefore, it is reasonable to infer that ovulating compared to non-ovulating women may prefer red, which represents a high-quality genetic signal.

On the other hand, according to the theory of female intra-sexual competition [42], women who are ovulating may also adopt derogatory or inhibitory strategies when competing with other women for a mate. These derogatory behaviors could increase a subjective sense of their own value and thus reduce feelings of threat [43]. Examples in the research include giving lower attractiveness ratings to other women [42] and less sharing of rewards or money with other women [44]. Men associated women's red clothing with sexual desirability [18,27], which can be a threatening sign for women competing for potential mates. Therefore, it is reasonable to infer that ovulating compared to non-ovulating females may devalue red, which is a more competitive threat signal. Although referee ratings are not typical of affiliative or competitive contexts, according to Elliot's CIC theory [16], the red effect is across contexts.

Our study is designed to fill a gap in the literature, namely the lack of research on gender and biological sex in relation to the red effect. In Experiment 1, the purpose of this study was to demonstrate whether red uniforms affects Wushu Routine practitioners' ratings for Wushu Routine athletes' position or movements, and what role the gender factor plays in this. The hypothesis of this study is that a red uniform, compared to a blue uniform, will enhance ratings of opposite-sex athletes but not same-sex athletes. In Experiment 2, the purpose of this study was to demonstrate whether the menstrual cycle of female Wushu Routine practitioners affects their ratings for position or movement of Wushu Routine athletes who wear red uniforms. The hypothesis of this study is that when in the ovulation phase, female practitioners will give higher ratings to male athletes wearing red and lower ratings to female athletes wearing red. Ratings of athletes wearing blue will not be affected by the phase of the female practitioners' menstrual cycle.

## 2. Experiment 1

### 2.1 Method

**2.1.1 Participants.** The participants were 144 college students (72 male, 72 female) who study in a School of Wushu. All participants had learned the specific skills of Wushu Routine and they are all Chinese. Among them, 96.5% of the subjects possessed the national athlete grade certificate of the Wushu Routine program, and the average length of practice of Wushu Routine was 6.88±3.85 (in years). They have all experienced Wushu Routine competitions and are more knowledgeable about Wushu Routine scoring. So, they have some ability to score Wushu Routine athletes in a referee role. The mean age was 20.7 years ($SD$ = 1.74), By self-report, all participants identified as heterosexual and did not have red-green color deficiency. The research was reviewed and approved by the university's Medical Ethics Committee. The participants provided their written informed consent to participate in this study. The start and end dates of the experiment are December 4, 2022 to December 30, 2022

**2.1.2 Design.** The experimental design was adopted a 2 (Color: red, blue) x 2 (Gender: male, female) ANOVA. According to the calculation of G*Power 3.1, the minimum number of participants required for ANOVA was 128 with a statistical power of 1 -β = 0.80, two-sided test α = 0.05, and effect size $f$ = 0.25. 144 Participants were randomly assigned to four groups based on uniform color (red vs. blue) and practitioners' gender (male vs. female) in a 2 x 2 between-subject design according to the method of random number table. There were 36 male athletes and 36 female athletes in the red uniform group, and 36 male athletes and 36 female athletes in the blue uniform group.

**2.1.3 Procedure.** The experiments are conducted in a university laboratory, which is quiet and comfortable, with sound and light insulation and a suitable temperature. The Monitor size is 19 inches with a resolution of 1024 x 768. E-prime software was used to conduct the experiments. A welcome screen described the study as being based on the participant's professional judgment, and explained that the participant would view twenty randomly presented photographs of Wushu routine, Changquan type athletes' movements. They were instructed to look at each photograph for 5s and then rate the quality of the each athlete's movements. After scoring was completed, they were told to proceed to the questionnaire, which was designed to probe for awareness of the manipulation. Participants were then debriefed and dismissed. The experiment was run individually, and the research assistants who conducted the sessions were unaware of the conditions or hypotheses.

**2.1.4 Materials.** Recruiting one male and one female Wushu Routine athlete. A separate group of participants (25males, 25 females) who did not participate in the experiment rated facial frontal photos for male and female athletes as being of average attractiveness, on a scale from 1 (not at all attractive) to 9 (extremely attractive). The mean attractiveness rating of the man was 6.02 ($SD$ = 1.26) and that of the woman was 5.57 ($SD$ = 1.33). The experimental materials were taken photographs of male and female athletes performing Changquan, wearing the same style of uniform by a photographer. The athletes in the photo were demonstrating the most representative movements in Changquan, including hand movements (fist, palm, and hook) and step movements (bow, horse, empty, slide and rest). Specifically, these movements include bow-step push palm, bow-step through palm, horse-step strike fist, rest-step push palm, rest-step show palm, empty-step show palm, slide-step through palm, empty-step push palm and longitudinal split. Experienced photographers used an EOS 600D camera to capture the athletes' movements. Another practitioner acted as the instructor during the shooting process. We selected 20 photos, 10 photos each of male and female athletes, all rehearsing similar actions. The photos of the athletes were processed using PS so that the red and blue colors were consistent in brightness and saturation. The hue could not be completely unified because the athletes' uniform would have folds and shadows when they performed their movements. Therefore, we made the athletes' uniform colors match the colors used in athletic competitions as much as possible. Ten practitioners of Wushu Routine who were not part of the formal study evaluated whether the athletes' uniform colors as presented in the photos were common competition uniform colors, and agreed that they were. After processing, The photos pixels are 5616*3744. The photos are shown in the paper, with the permission of both athletes. The final one group of photos (The photos in the experiment are not mosaic) used in the experiment are shown in Figs 1–4.

**2.1.5 Measures.** Movement quality evaluation: "Please rate the quality of the athlete's movements in the pictures from the perspective of a referee." Responses were given using a 9-point Likert scale from 1 (not at all good) to 9 (extremely good).

Awareness probe: Participants' awareness of the purpose of the study was assessed using four items. Participants were asked to indicate how much their rating was influenced by "the coordination, stretching, and strength of movement"; "athletes' essence, qi, and spirit"; "athletes' uniform color"; and "athletes' attractive appearance."

## 2.2 Results

We conducted two analyses on the movement quality evaluations, one for ratings of male athletes, and one for ratings of female athletes. Each analysis was a 2 (Color: red, blue) x 2 (Gender: male, female) ANOVA, with the movement quality evaluation score as the dependent measure.

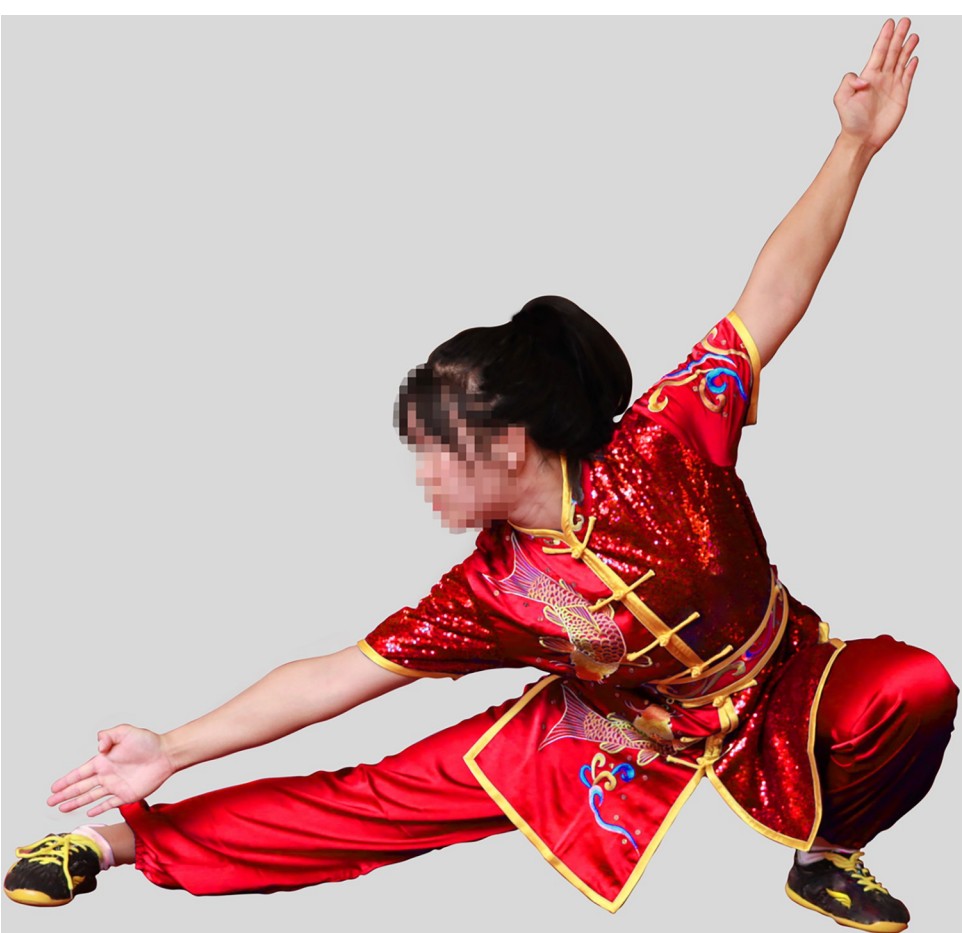

**Fig 1. Female athlete in red uniform (slide-step through palm).**

The first analysis, when the target athletes were all male, showed a significant main effect of uniform color, $F(1, 140) = 9.903$, $p = .002$, $\eta^2 = .066$, with significantly higher scores for male athletes wearing a red uniform ($M = 7.49$, $SD = 0.91$) than for male athletes wearing a blue uniform ($M = 6.97$, $SD = 1.07$). The main effect of the practitioner gender was not significant, $F(1, 140) = 1.246$, $p = .266$, $\eta^2 = .009$. The interaction between uniform color and practitioner gender was not significant, $F(1, 140) = 0.355$, $p = .552$, $\eta^2 = .003$. See Fig 5.

The second analysis, when the target athletes were female, showed a significant main effect of uniform color, $F(1, 140) = 6.249$, $p = .014$, $\eta^2 = .043$, with significantly higher scores for female athletes wearing red ($M = 6.23$, $SD = 1.10$) than for female athletes wearing blue ($M = 5.76$, $SD = 1.20$). The main effect of practitioner gender was not significant, $F(1, 140) = 0.359$, $p = .550$, $\eta^2 = .003$. The interaction between uniform color and practitioner gender was significant, $F(1, 140) = 4.188$, $p = .043$, $\eta^2 = .029$. A simple effects analysis of the interaction found that male practitioners gave significantly higher ratings to female athletes wearing a red uniform ($M = 6.48$, $SD = 1.02$) than to female athletes wearing a blue uniform ($M = 5.62$, $SD = 1.41$), $p = .002$, $\eta^2 = .069$. Female practitioners gave similar ratings of female athletes wearing blue ($M = 5.89$, $SD = 0.83$) and female athletes wearing red ($SM = 5.98$, $SD = 1.13$), $p = .749$. See Fig 6.

We then compared the four items on the awareness probe in relation to practitioners' ratings: For the item on coordination, strength, and strength, $M = 8.17$ ($SD = 0.78$); For the item

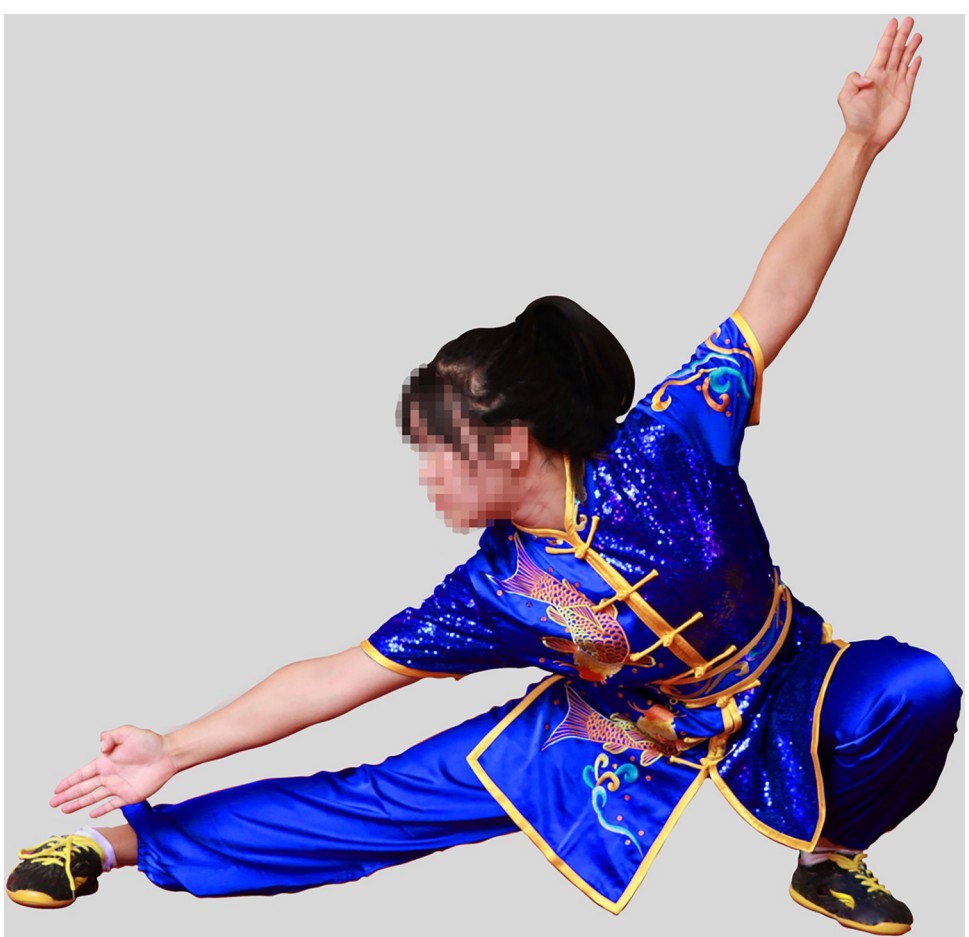

**Fig 2. Female athlete in blue uniform (slide-step through palm).**

on essence, qi, and spirit, $M = 8.13$ ($SD = 1.12$); For the item on attractive appearance, $M = 5.60$ ($SD = 1.59$); For the item on uniform color, $M = 5.81$ ($SD = 1.98$). Post hoc tests showed that the ratings for uniform color were significantly lower than the ratings for coordination, strength, and strength ($p < .001$); essence, qi, and spirit ($p < .001$); but not significantly different than the ratings for attractive appearance ($p > .05$).

The results of Experiment 1 supported part of our hypothesis. When when the target athlete was male, the red uniform improved the practitioners' ratings compared to blue uniform, but practitioners' gender did not affect the ratings. When the target athlete was female, practitioners' ratings were higher when the uniform was red than blue, and the red effect was more pronounced when the practitioner was male. Despite the fact that practitioners' ratings were influenced by the color red, they rated the uniform color as having the least influence on their ratings, compared to the components of the routine. The practitioners appeared not to realize that uniform color affect their ratings.

## 3. Experiment 2

### 3.1 Method

**3.1.1 Participants.**   The participants were 52 female college students from the School of Wushu, all participants were studying the specific skills of Wushu Routine and they are all

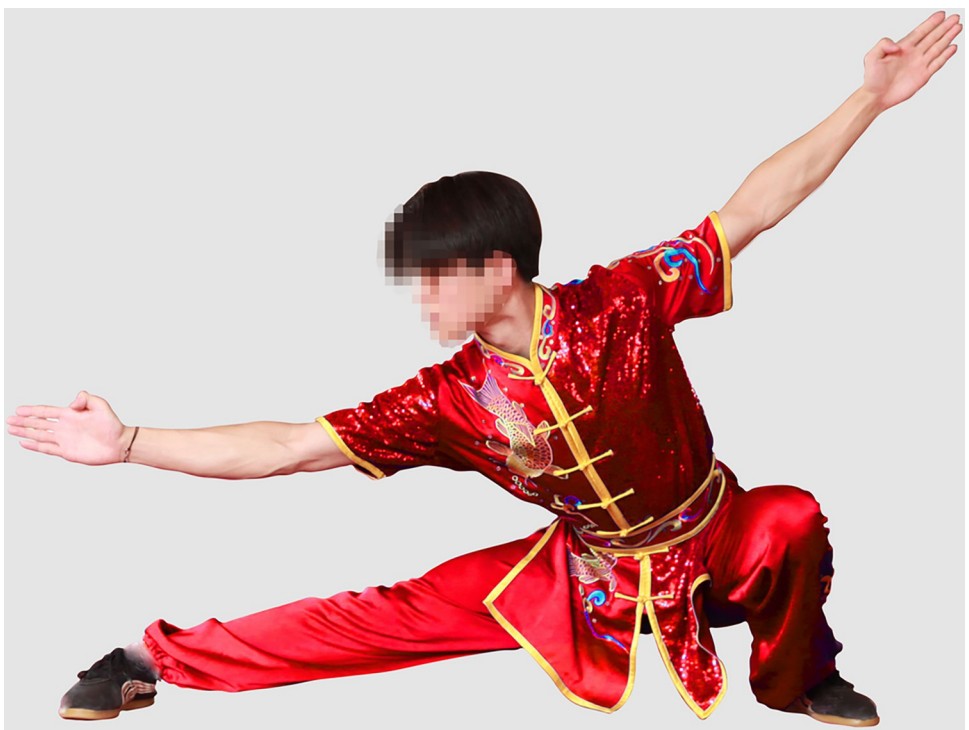

**Fig 3. Male athlete in red uniform (slide-step through palm).**

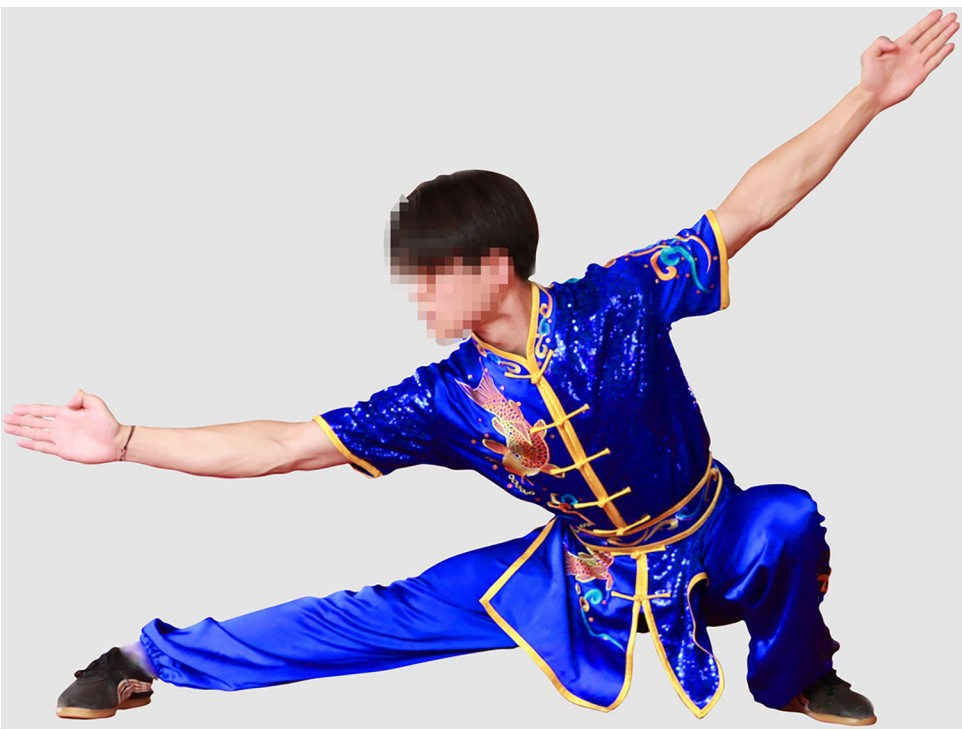

**Fig 4. Male athlete in blue uniform (slide-step through palm).**

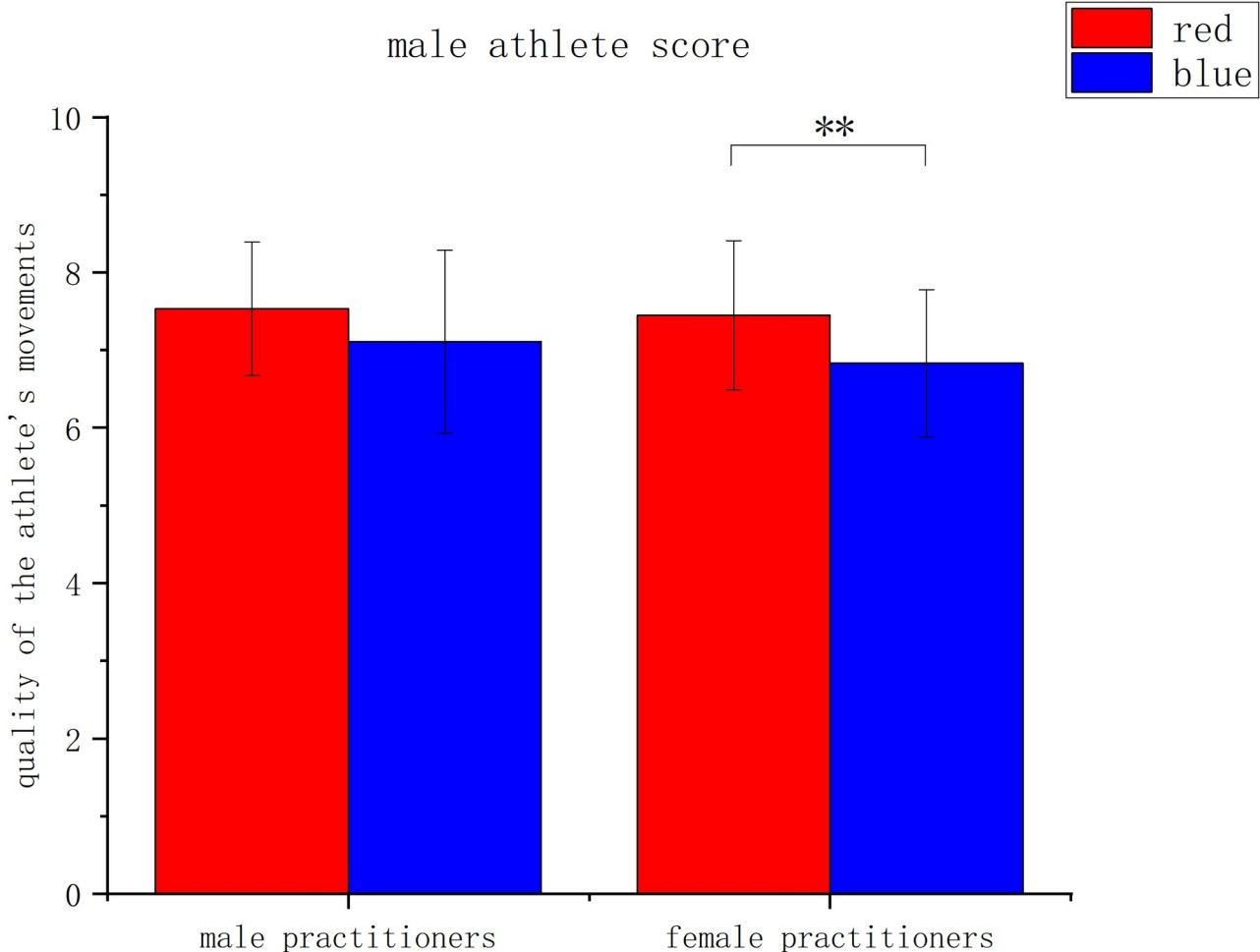

**Fig 5. Assessment of the movements of male athletes considering the color of the athlete's uniform and the sex of the evaluator.** Note: * indicates $P < .05$, ** indicates $P < .01$, *** indicates $P < .001$, $P$ values indicate the results of independent sample t-tests).

Chinese. Among them, 94.2% of the subjects possessed the national athlete grade certificate of the Wushu Routine program, and the average length of practice of Wushu Routine was 6.25 ±3.16 (in years). They have all experienced Wushu Routine competitions and are more knowledgeable about Wushu Routine scoring. So, they have some ability to score Wushu Routine athletes in a referee role. The mean age was 20.03 years ($SD = 0.98$). By self-report, all participants identified as heterosexual, had no color blindness, and within the last six months had a stable menstrual cycle of 28–30 days. The backward-counting method [45] was used to determine the two phases of the menstrual cycle during which women participated in the experiment: ovulation (14–19 days backwards from the first day of the estimated next menstruation), and non-ovulation (to avoid self-reported inaccuracy in backward counting, participants were asked to do the rating task at least three days after the end of the estimated ovulation period). The research was reviewed and approved by the university's Medical Ethics Committee. The participants provided their written informed consent to participate in this study. The start and end dates of the experiment are February 25, 2023 to April 29, 2023

**3.1.2 Design, procedure, materials, and measures.** We used within-subjects design to conducted experiment separately for the blue and red groups. According to the calculation of

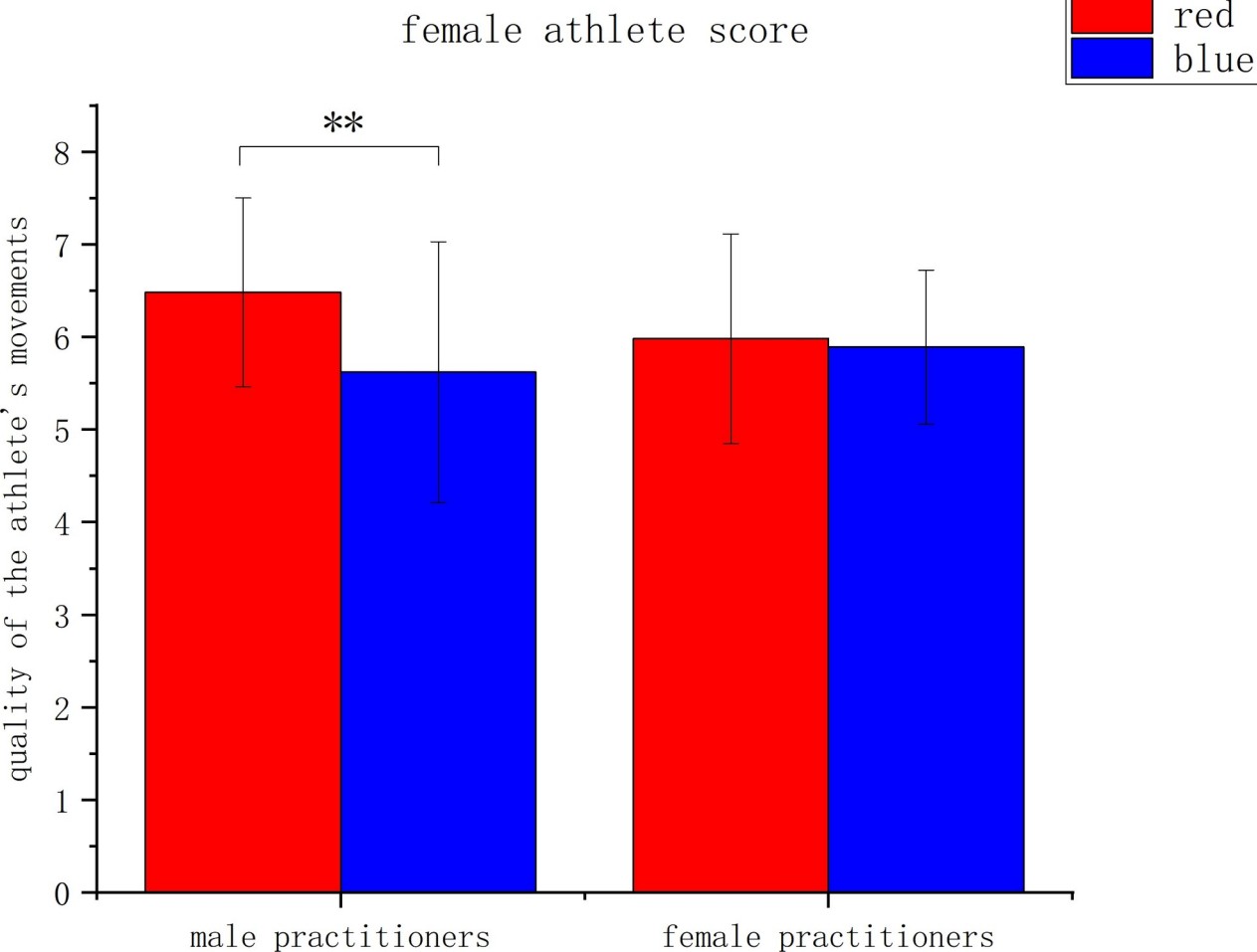

**Fig 6. Assessment of the movements of female athletes considering the color of the athlete's uniform and the sex of the evaluator.** Note: * indicates $P < .05$, ** indicates $P < .01$, *** indicates $P < .001$, $P$ values indicate the results of independent sample t-tests).

G*Power 3.1, the minimum number of participants required for Paired samples t-test was 15 with a statistical power of $1 - \beta = 0.80$, two-sided test $\alpha = 0.05$, and effect size $d = 0.80$. The female practitioners were randomly assigned to two groups according to the method of random number table (26 persons per group) and made ratings of the athletes once during the ovulation phase and once during the non-ovulation phase. The order in which the tasks were completed was balanced between subjects: Half of the females in the blue group were tested first during ovulation and then during non-ovulation; the other half were tested first during non-ovulation and then during ovulation. The red group did the same arrangement. Finally, After a follow-up investigation, excluding five participants whose menstrual cycles did not remain stable after the experiment, the total number of participants in the red uniform group was 24, and the total number of participants in the blue uniform group was 23.

The procedure, materials, and measures were the same as in Experiment 1.

### 3.2 Results

The data collected during each woman's ovulatory and non-ovulatory phases were compared using paired samples *t* tests.

When female practitioners scored male athletes wearing red, the scores for ovulation ($M = 7.56$, $SD = 0.89$) were significantly higher than the scores for non-ovulation ($M = 7.21$, $SD = 0.92$), $t(23) = 2.382$, $p < .05$. When female practitioners scored female athletes wearing a red uniform, the scores for ovulation ($M = 5.86$, $SD = 1.39$) were not significantly different from the scores for non-ovulation ($M = 5.95$, $SD = 1.32$), $t(23) = 0.472$, $p > .05$. When female practitioners scored male athletes wearing blue, the scores for ovulation ($M = 6.93$, $SD = 1.06$) were not significantly different from the scores for non-ovulation ($M = 6.85$, $SD = 1.08$), $t(22) = 0.473$, $p > .05$. When the female practitioners scored the female athletes wearing blue, the scores for ovulation ($M = 5.87$, $SD = 1.16$) were not significantly different from the scores for non-ovulation ($M = 5.82$, $SD = 1.39$), $t(22) = 0.463$, $p > .05$.

Experiment 2 demonstrated that female practitioners gave higher ratings to male athletes wearing a red uniform during ovulation (vs. non-ovulation). In contrast, menstrual cycle fluctuations had no effect on female practitioners' ratings of female athletes wearing a red uniform, or their ratings of athletes of either gender wearing a blue uniform.

## 4. Discussion

The color red has been shown to affect practitioner judgement but exploration in artistic sports is rare. This study investigated whether the color red affects ratings made of athletes' position or movement of Wushu Routine. College students who were studying Wushu Routine were asked to take the role of referee and rate the position or movement of athletes wearing a red or blue uniform. We also tested whether the gender of the practitioner and athlete (Experiment 1) or the ovulation phase of female practitioners' menstrual cycle (Experiment 2) affected the practitioners' ratings. Past studies have demonstrated the existence of the red effect, with contextual factors playing a key role in the red effect. The results confirmed a red effect in the context of Wushu Routine rating and showed that this effect was especially strong when male practitioners rated female athletes. There was minimal evidence of variation in the red effect based on whether or not the rater was in the ovulation phase. The results also confirmed ovulating female practitioners will rate male athletes wearing red clothing higher.

Athletes wearing a red uniform in Wushu Routine received higher ratings from both male and female practitioners. One explanation has to do with the biologically based responses to the color red. Red may enhance practitioners' ratings by presenting biologically based signals of aggressiveness and dominance [28,46,47]. The effect of the red uniform is in line with the nature of Wushu, which portrays aggression, albeit in a stylized, aesthetically pleasing form. The results of our study supported this possibility, although based on their responses to the awareness probe, the practitioners appeared unaware than uniform color affected their ratings.

However, practitioners' psychological association between red and aggression may vary based on whether or not aggression is a key component of the sport practitioners might judge athletes who in red uniform more favorably in allows display of aggresive sports, such as taekwondo [13]. By contrast, practitioners might judge athletes who in red uniform unfavorably in sports of need to constrain aggressiveness through rules, such as soccer [15]. So, in sports where physical aggression is expected, red uniform may influence the practitioner to make favorable judges; while in sports where physical aggression is expected red uniform may influence the practitioner to make unfavorable judges. This is an avenue for future research.

The red effect in Wushu ratings may also be a function of the special psychological meaning of red in Chinese culture. Chinese people have a strong affinity for the color red. Chinese people often attribute the sentiment for the color red to worshipping the sun and fire when living in primitive societies [48]. In ancient China, red was a symbol of wealth and power; a home with a "red door" indicated power and nobility. In the Song Dynasty, only those in power

could use red [49]. Red also represents joy and peace in folklore, as red is used in costumes and scenes of classical Chinese weddings to highlight the festivity. Couplets printed in black ink on red paper are used to express blessings during the Chinese New Year, the most important festival in China. Additionally, the base color of the Chinese five-star red flag is red. Red has clear psychological meaning in the minds of Chinese people. Wushu Routine practitioners may rate athletes wearing red more favorably in this cultural context.

Contrary to expectations, there was no interaction between practitioners' gender and uniform color in predicting ratings when the target athlete was male. The reason may be that red uniform has increased the rating of female practitioners on male athletes, and it has also increased the rating of male practitioners on male athletes. In terms of female evaluation of male, studies showed red-attractiveness effect exist in women's evaluations of men [22]. The evolutionary argument is that this gender difference may be due to women's association of red color with health [29], dominance [28] and status [22]. These characteristics might serve as indicators of the quality of potential male mates' genetic material. In terms of male evaluation of male, Ioan et al. found that men and women differed in their responses to red messages, and asserted that over the course of human evolution [50], males came to associate other males' demonstrations of red with aggression and dominance. As discussed above, aggresion is in line with the nature of Wushu. Thus, the red uniform may enhance female practitioners' ratings of male athletes. With the two influences at work, the difference in ratings of male athletes based on the practitioner's gender may not be significant.

Consistent with our expectations, when the target athletes were female, there was an interaction between practitioner gender and uniform color; specifically, male practitioners (vs. female practitioners) gave higher ratings to female athletes wearing a red uniform (vs. a blue uniform). The results are consistent with previous studies on the female red-attractiveness effect. Evolutionary psychologists would suggest that female nonhuman primates' red skin is a sexual signal of fertility. The redder the female's skin, the more likely the female is to reproduce [24–26,51], and males will gaze longer at the reddened torso of females [52]. Studies on the red effect in humans have also shown that men rate women wearing a red clothing more positively, and that sexual receptivity mediates the relationship between redness and attraction [27].

Thus, research on both nonhuman and human primates suggests that there is a relationship between redness and reproduction. However, previous studies and the current study showed that humans are unaware of the redness effect [22,53]; the response to red may be unconscious and automatic. There is also cross-cultural evidence that the red-sex association has been found not only in Western cultural contexts, but also in isolated traditional social groups [54]. Thus, it is possible that the red effect is a psychological adaptation that evolved with humans. In addition, it has been argued that the effect of red on men's ratings of women's attractiveness may be more robust than women's ratings of men's attractiveness [29], because men's evaluations of women are less disturbed by other factors such as women's emotional expression [55] or conception risk [56], and the perceptions of the women's aggression [28] or anger [57].

Our results showed that when female practitioners were in the ovulation phase (vs. non-ovulation phase) of their menstrual cycle, they gave higher ratings to male athletes who wore a red uniform. By contrast, there are no difference in female practitioners rating to male athletes who wore blue at their different menstrual cycles. Women at times of ovulation phase have been shown to report a stronger sexual desire for a man surrounded by a red, relative to gray, border [56]. This suggests that redness exists as an attraction associated with the motivation to reproduce, and that this attraction exists in neutral contexts outside of affiliative contexts (e.g., Wushu Routine scoring). This may be due to both biological evolution and social learning.

In nonhuman primates, the red coloration of the male body surface (for example in mandrills, whose face, buttocks, and genitals are red) is linked to high status, dominance, and

health [58]. Researchers who have studied primates such as geladas and macaques have come to similar conclusions [26,59]. This may be due to the effect of testosterone on the social status and competitive advantage in males [60]. Testosterone production induces changes in the peripheral blood vessels, which in turn cause the skin to redden [61]. Higher levels of both carotenoids and oxygenated hemoglobin also cause the skin of animals to become redder [62,63]. The role of testosterone, carotenoids, and oxygenated hemoglobin is similar for human and nonhuman primates, with the key differences being the unique survival environments and the social culture that humans have constructed. For humans, there are fewer and fewer scenarios that require direct physical competition for social status, and the human body surface (especially in relation to reproduction and lactation) is obscured by clothing.

However, psychological mechanisms based on a long evolutionary history may still influence human beings and extend to social and cultural symbols. In ancient China, there were revered red dynasties, such as the Zhou, Han, Sui, and Song dynasties, during which red carried the meaning of wealth and prominent status. The reverence for red was especially strong during the Song dynasty, when red had an aesthetic function but also was used as evidence of power. The imperial power of the Song Dynasty even monopolized certain sources of red color, such as red clay, and it was a crime for ordinary adult men to wear red clothes, or for ordinary people to use red furniture [49]. In modern times, red can still convey status. For example, "go red" means a great social status or reputation enhancement, "red people" also means a display of social stutus in China. In addition,the red carpet, and the red tie as a power symbol.

The color red has a unique value for women based on the interaction of biological evolution and acquired learning. Elliot et al. showed that status perceptions were responsible for women's preference for men who wore red or were in a red background [22]. Stephen et al. asked women in their experiment to find ways to make the target men look more dominant, and women tended to leave more red lipstick on the target men's faces [28]. Thorstenson et al. showed that male facial redness positively influenced heterosexual females' ratings of attractiveness, sexual attractiveness, and health [29]. In addition, perceived health was shown to mediate the effect of men's facial redness on women's perceptions of their attractiveness. A man's high status, dominance, and health are highly valued conditions for women. Status induction has been shown to increase women's ratings of male attractiveness and desirability [64]. Women have been shown to describe social status as a necessity, not luxury, in choosing a mate or partner [65,66]. In addition, various indicators of male status are positive predictors of reproductive success, across cultures and types of societal structures [67]. This may be because a man's high status confers more resources to a female partner, increasing the likelihood that the woman will pass healthy genes to the next generation.

The female preference for good genes may be moderated by the phase of her menstrual cycle. Females have the opportunity to benefit from good genes only during ovulation [41], because good genes contribute to reproduction and children's healthy development. As previously mentioned,for women, the meaning of red in men is associated with quality genes. Therefore, males who wear a red clothing may be more likely to induce preference in ovulatory females. According to CIC theory, color with unique meaning automatically influences people's evaluative processes. An association of red with quality genes provides a possible explanation for our finding that female practitioners who were ovulating rated male athletes wearing a red vs. blue uniform more favorably.

We expected that female practitioners were in the ovulation phase (vs. non-ovulation phase) will gave lower ratings to female athletes who wore a red uniform. However, the results did not support this hypothesis. Female practitioners who were ovulating (vs. not) gave similar ratings to female athletes wearing red and blue uniforms. A study also showed that ovulating

compared to non-ovulating women showed no differences in behavior or perceptions of female athletes wearing a red clothing [68].

There are several possible reasons for the lack of a red effect when female practitioners rated female athletes. Firstly, in the Wushu Routine scoring, female practitioners and female athletes have no competition for males, and thus there may be a lack of external motivation for the practitioner to demean the athlete, even if the practitioner is ovulating. Maner et al. showed that when female participants' motivation for mate seeking was induced, attractive photos of the opposite sex (potential mates) prolonged the participants' responses; when motivation for mate guarding was induced, same-sex photos (competitors) prolonged the participants' responses [69]. This suggests that physiological and psychological factors influence how women perceive others women, but external situational factors also influence these perception.

Secondly, if the photo shows a woman wearing relatively formal clothes with little skin exposure, the red clothing reduces other women's ratings of her attractiveness [19]. In one study, when women viewed a photo of a sexy woman in revealing clothes, they were more likely to derogate the sexual fidelity of the woman in the photograph, regardless of whether there was induced competition; by contrast, there was no effect on the evaluation of the woman's financial resources [70]. This suggests that the choice of the experimental photographs is related to the red effect when women are rating other women. The female athletes chosen for this study were in formal Wushu Routine performance uniform and therefore may not have caused ovulating practitioners to score the female athletes less favorably.

It is worth noting that in the scoring of Wushu Routine athlete, there is a multiplicity of factors that referees need to consider, uniform color is only one of the influencing factors. In addition, red effect may be subtle, it is possible that the effect of red uniform will only be exerted if the athletes are at a comparable level and have similar styles, as controlled for in the experiment. Thus, although the red effect plays a role in judges' scoring of Wushu Routine athletes, this role should not be overestimated.

## 5. Limitations

This study has several limitations. The first lies in the problem of small sample size. Relatively few students major in Wushu Routine, limiting our ability to create a larger sample. Second, although the participants had learned Wushu Routine through their Practical and theoretical learning, their ratings are likely to differ from those made by experienced referees, this study did not test whether the ratings of Wushu Routine practitioners were consistent with the ratings of experienced Wushu Routine referees, future studies could verify the reliability and reproducibility of experimental results with experienced referees as participants. Third, although the stimulus photos were processed to show uniformity in the brightness and saturation of the red and blue garments, we did not alter hue as a way to increase the perception of naturalness and authenticity. Doing so might have improved the study's ecological validity. Fourth, in Experiment 2, a self-report method was used to determine ovulation and non-ovulation, and although this method has been used in most research on the female menstrual cycle, there may still exist bias. Finally, The Chinese Wushu Routine athletes and Chinese Wushu Routine practitioners were selected for this study, while the connotation of the color red varies in different countries, and the applicability of the conclusions drawn in other cultural contexts needs to be further explored.

## 6. Conclusions

This study documented a red effect in scores given to athletes in Wushu Routine (Changquan), a sport that has strong Chinese artistic characteristics. Experiment 1 showed that both male

and female Wushu Routine athletes wearing red uniform (compared to the blue uniform) resulted in higher ratings from Wushu Routine practitioners for their position or movement, but gender moderates this red effect only when male practitioners rated female athletes. Experiment 2 showed that female Wushu Routine practitioners' menstrual cycle fluctuations do not affect their ratings of the position or movement of female athletes in red and blue uniform and male athletes in blue uniform, only male athletes in red uniform, and specifically, they rate male athletes in red uniform higher during ovulation phase of their cycle.

## Supporting information

**S1 File.**
(ZIP)

**S2 File.**
(ZIP)

**S3 File.**
(ZIP)

## Acknowledgments

We would like to thank all the participants in these experiments and the members of our research group for their assistance.

## Author Contributions

**Conceptualization:** Xiaobin Hong.

**Data curation:** Jinkun Li, Jingmin Zhang, Shuo Tao.

**Formal analysis:** Jinkun Li, Shuo Tao.

**Investigation:** Jinkun Li, Jingmin Zhang, Xiaoying Zeng.

**Methodology:** Jinkun Li, Rong Zou.

**Project administration:** Jingmin Zhang, Xiaoying Zeng.

**Resources:** Xiaobin Hong.

**Software:** Jinkun Li, Jingmin Zhang, Shuo Tao.

**Supervision:** Rong Zou.

**Visualization:** Xiaobin Hong.

**Writing – original draft:** Jinkun Li.

**Writing – review & editing:** Xiaobin Hong.

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
