## [Decision Letter · Decision Letter 0]

23 Nov 2023

PONE-D-23-24644The effect of red uniform on judging Wushu Routine athletes' sport performancePLOS ONE

Dear Dr. Hong,

Thank you for submitting your manuscript to PLOS ONE. After careful consideration, we feel that it has merit but does not fully meet PLOS ONE’s publication criteria as it currently stands. Therefore, we invite you to submit a revised version of the manuscript that addresses the points raised during the review process.

**ACADEMIC EDITOR: **

The reviewers pointed out important aspects to be answered, as well as suggestions. If you choose to continue with the submission process, please send a response for each point indicated by the reviewers.

We look forward to receiving your revised manuscript.

Kind regards,

Leonardo Vidal Andreato, PhD

Academic Editor

PLOS ONE

Reviewers' comments:

Reviewer's Responses to Questions

**Comments to the Author**

1. Is the manuscript technically sound, and do the data support the conclusions?

Reviewer #1: Partly

Reviewer #2: Yes

2. Has the statistical analysis been performed appropriately and rigorously? 

Reviewer #1: Yes

Reviewer #2: Yes

3. Have the authors made all data underlying the findings in their manuscript fully available?

Reviewer #1: Yes

Reviewer #2: Yes

4. Is the manuscript presented in an intelligible fashion and written in standard English?

Reviewer #1: Yes

Reviewer #2: Yes

5. Review Comments to the Author

Reviewer #1: The study is interesting and provides relevant information for the study area, as well as for improving refereeing in Wushu, combat sports and other sports.

Some points should be considered, for example, the judging was carried out by Wushu practitioners and not by experienced referees. Another point, the analysis was carried out using photos of a position used in Wushu routines, but the analysis of performance judgment in competitions is much more complex.

Suggestion for the study:

In the title:

Instead of "sports performance", which is complex, it should be "Effect of the red uniform on the judgment of a position or movement used in Wushu routines, evaluated by practitioners of the modality."

In the abstract:

The authors describe that the study evaluates whether the color of the uniform can affect the evaluation of "referees", however the study evaluates whether the color of the uniform affects the evaluation of "wushu practitioners". It is suggested to adjust.

In the introduction:

It is suggested, at the end of introduction, before presenting hypotheses, to make the objectives of each of the experiments clearly.

In the Method:

Regarding the number of participants, was any type of sample calculation carried out? If it has not been done, add the information in the study limitations.

Would it be possible to include the participants' degree (level of knowledge) in Wushu, as well as the average time spent practicing in the modality?

The authors present just 4 photos, however, in the methods it is described that the participants evaluated 20 photos, it is suggested to place all the photos used in annex. It would be interesting to describe which positions or movements were evaluated.

Was any reproducibility assessment of the grades awarded carried out? Was a pilot study carried out to ensure the reproducibility of the assessments?

As in experiment 1, describe the degree (level of knowledge) and practice time in Wushu of the participants in study 2.

In the discussion:

In the discussion it is important to emphasize that sporting performance and referees' judgment is multifactorial, and the variable analyzed in the present study is one of these factors.

On page 21 of the study (Page 27 of the PDF), in line 466, where it says "imitations" insert "limitations".

In conclusion:

As in the title, the conclusion should describe that the effects of color (Red and blue) and the menstrual cycle (women in relation to ovulation vs. non-ovulation), are about judging movements or positions used in Wushu (Changquan), carried out by practitioners of the modality.

In the figures:

Improve the description of figures. Insert titles in the figures, indicate where significant differences were observed (e.g. *). The figures must be self-explanatory.

Reviewer #2: Dear Authors

I believe the content of this article holds significant promise for publication. However, I would like to offer some suggestions to improve the quality of the manuscript.

Abstract:

I suggest inserting numerical data from the results in the abstract. For example, in “The results of Experiment 1 showed that both male and female referees gave higher ratings to athletes wearing red uniforms, but the red effect was especially strong when male referees rated female athletes.” How much would that be?

Methods

I suggest specifying Wushu practice time in numerical data (mean and standard deviation) in both Experiment 1 and Experiment 2. And did they have experience refereeing competitions (local, national or international)?

As you commented in the discussion about the psychological meaning of the color red in Chinese culture, I suggest identifying in the methods whether the study participants are Chinese.

Results

I suggest adding the p-value of the statistically significant difference in Figures 5 and 6.

Discussion

I suggest indicating as a limitation of the study the need to carry out this model of studies with referees from different nationalities. In this way, it could be identified whether the effect of the red color also occurs with referees from other countries, where red does not have the same meaning that is attributed to it in China. I also suggest reading an article that analyzed the influence of the blue color of the judo uniform on victory in competitions in Brazil (available in https://doi.org/10.2466/30.PMS.120v15x2), perhaps it can contribute to the discussion of the data.

6. PLOS authors have the option to publish the peer review history of their article (what does this mean?). If published, this will include your full peer review and any attached files.

Reviewer #1: No

Reviewer #2: No

---

## [Author Response · Author response to Decision Letter 0]

18 Dec 2023

We feel great thanks for your professional review work on our article. As you are concerned, there are several problems that need to be addressed. According to your nice suggestions, we have made extensive corrections to our previous draft, the detailed corrections are listed below.

Dear academic editor:

Thank you for your valuable suggestion on the paper! We have made the following amendments in response to your suggestions

1.Please ensure that your manuscript meets PLOS ONE's style requirements, including those for file naming.

Thank you for your advice. We refer to the pdf file given by the journal to make appropriate adjustments to the font.

2.If you’ve not already done so, consider depositing your raw data in a repository to ensure your work is read, appreciated and cited by the largest possible audience.

Thank you for your advice. We have uploaded our raw data as well as the results of our analysis to "Zenodo", specifically "10.5281/zenodo.10325917".

3.We note that the grant information you provided in the ‘Funding Information’ and ‘Financial Disclosure’ sections do not match.When you resubmit, please ensure that you provide the correct grant numbers for the awards you received for your study in the ‘Funding Information’ section. 

Thanks for the heads up. When we resubmit, we will provide the correct grant numbers for the awards.

4.Upon re-submitting your revised manuscript, please upload your study’s minimal underlying data set as either Supporting Information files or to a stable. 

Thank you for your advice. We have uploaded our raw data as well as the results of our analysis to "Zenodo", specifically "10.5281/zenodo.10325917".

5.Please ensure that you have an ORCID iD and that it is validated in Editorial Manager. 

Thanks for the heads up. We have validated the corresponding author's ORCID iD in Editorial Manage

Dear expert 1:

We sincerely appreciate the valuable suggestions you have given us! In response to your suggestions, we have made the appropriate changes.

1.In the title: Instead of "sports performance", which is complex, it should be "Effect of the red uniform on the judgment of a position or movement used in Wushu routines, evaluated by practitioners of the modality."

Thank you for your suggestion, we agree with you! We changed the title to "Effect of the red uniform on the judgment of a position or movement used in Wushu routines, evaluated by practitioners of the modality." And we have replaced "referees" with "practitioners", replaced “performance” with “position or movement” in the corresponding section of the article.

2.In the abstract: The authors describe that the study evaluates whether the color of the uniform can affect the evaluation of "referees", however the study evaluates whether the color of the uniform affects the evaluation of "wushu practitioners". It is suggested to adjust.

Thank you for your suggestion, we agree with you! We have replaced "referees" with "practitioners" in the abstract, and have made the corresponding substitution in the article.

3.In the introduction: It is suggested, at the end of introduction, before presenting hypotheses, to make the objectives of each of the experiments clear.

Thank you for your suggestion, we agree with you! In the introduction section, the purpose of each experiment was further clarified before presenting hypotheses.Specifically in lines 121-123 and 125-127.

4.In the Method: Regarding the number of participants, was any type of sample calculation carried out? If it has not been done, add the information in the study limitations.

Thank you for your suggestion, we agree with you! We pre-calculated the number of subjects for each experiment using G*Power 3.1, and added the methods section in lines 186-188 and 311-313.

5.Would it be possible to include the participants' degree (level of knowledge) in Wushu, as well as the average time spent practicing in the modality?

Thank you for your suggestion. We counted and supplemented the participants' athlete levels as well as the number of years of practice in lines 176-179.

6.The authors present just 4 photos, however, in the methods it is described that the participants evaluated 20 photos, it is suggested to place all the photos used in annex. It would be interesting to describe which positions or movements were evaluated.

Thank you for your suggestion. We will add 20 photographs of male and female from the experimental material to the appendix and described the actions shown in each photo.

7.Was any reproducibility assessment of the grades awarded carried out? Was a pilot study carried out to ensure the reproducibility of the assessments?

Thank you for your suggestion. We assessed this only for experimental materials, but did not conduct a reproducibility assessment and pilot experiment on participants' ratings of athletes. Thank you for your suggestions on the design shortcomings of this article.

8.As in experiment 1, describe the degree (level of knowledge) and practice time in Wushu of the participants in study 2.

Thank you for your suggestion. We counted and supplemented the participants' athlete levels as well as the number of years of practice in lines 175-179 and 295-298.

9.In the discussion it is important to emphasize that sporting performance and referees' 

judgment is multifactorial, and the variable analyzed in the present study is one of these 

factors.

Thank you for your suggestion, we agree with you! In the discussion section, we emphasize that although red effect plays a role in judges' scoring of Wushu Routine athletes, this role should not be overestimated in lines 487-492.

10.On page 21 of the study (Page 27 of the PDF), in line 466, where it says "imitations" 

insert "limitations".

Thank you for your suggestion, we agree with you! We insert “limitations” in line 493.

11.In conclusion: As in the title, the conclusion should describe that the effects of color (Red and blue) and the menstrual cycle (women in relation to ovulation vs. non-ovulation), are about judging movements or positions used in Wushu (Changquan), carried out by practitioners of the modality.

Thank you for your suggestion, we agree with you! Based on your suggestion, we have described the conclusion in more detail in lines 508-515.

12.In the figures: Improve the description of figures. Insert titles in the figures, indicate where significant differences were observed (e.g. *). The figures must be self-explanatory.

Thank you for your suggestion, we agree with you! Based on your suggestion,we provide a more detailed description of the figures in lines 255-258 and 271-273, and insert titles in the figures, significant differences are indicated by “*”. 

Dear expert 2:

We sincerely appreciate the valuable suggestions you have given us! In response to your suggestions, we have made the appropriate changes.

1.I suggest inserting numerical data from the results in the abstract. For example, in “The results of Experiment 1 showed that both male and female referees gave higher ratings to athletes wearing red uniforms, but the red effect was especially strong when male referees rated female athletes.” How much would that be?

Thank you for your suggestion, we agree with you! We have added relevant data to the abstract to illustrate the red effect more clearly.

2.I suggest specifying Wushu practice time in numerical data (mean and standard deviation) in both Experiment 1 and Experiment 2. And did they have experience refereeing competitions (local, national or international)?

Thank you for your suggestion, we agree with you! We counted and supplemented the participants' athlete levels as well as the number of years of practice. But, despite their knowledge of refereeing, they have not formally begun to work in refereeing-related jobs. So we have replaced "referees" with "practitioners" in the abstract, and have made the corresponding substitution in the article.

3.As you commented in the discussion about the psychological meaning of the color red in Chinese culture, I suggest identifying in the methods whether the study participants are Chinese.

Thank you for your suggestion, we agree with you! All participants are Chinese and added the participant's nationality to the method section.

4.I suggest adding the p-value of the statistically significant difference in Figures 5 and 6.

Thank you for your suggestion, we agree with you! We added the p-value of the statistically significant difference in Figures 5 and 6.

5.I suggest indicating as a limitation of the study the need to carry out this model of studies with referees from different nationalities. In this way, it could be identified whether the effect of the red color also occurs with referees from other countries, where red does not have the same meaning that is attributed to it in China.

Thank you for your suggestion, we agree with you! It is essential to consider the influence of cultural factors on the red effect, which is one of the limitations of this study. We thus add this limitation to the “limitations” section in lines 503-505.

6.I also suggest reading an article that analyzed the influence of the blue color of the judo uniform on victory in competitions in Brazil, perhaps it can contribute to the discussion of the data.

Thank you for your suggestion. I have carefully read Julio's study (Blue judogi may bias competitive performance when seeding system is not used: sex, age, and level of competition effects) as well as other studies (Sporting contests: seeing red? Putting sportswear in context; Blue judogis may bias competition outcomes) that have found that a higher probability of an athlete's winning a combat wearing blue judogi. It is indeed interesting to explore the blue effect, but this blue effect seems to be controversial (no effect of blue on winning contests in judo), and this type of study is limited to only one sport, in addition to the fact that there is not enough experimental research to prove that the blue effect exists, thus we didn't include the blue effect in the discussion. That's why we didn't include Julio's paper in the discussion portion of this paper, I hope you understand that.

---

## [Decision Letter · Decision Letter 1]

16 Feb 2024

PONE-D-23-24644R1Effect of the red uniform on the judgment of position or movement used in Wushu Routine, evaluated by practitioners of the modalityPLOS ONE

Dear Dr. Hong,

Thank you for submitting your manuscript to PLOS ONE. After careful consideration, we feel that it has merit but does not fully meet PLOS ONE’s publication criteria as it currently stands. Therefore, we invite you to submit a revised version of the manuscript that addresses the points raised during the review process.

**ACADEMIC EDITOR: **Dear author, as you can see, the reviewers approved the changes made. However, a few details still need to be adjusted. Please, adjust it and highlight these changes in the new submission.

We look forward to receiving your revised manuscript.

Kind regards,

Leonardo Vidal Andreato, PhD

Academic Editor

PLOS ONE

Journal Requirements:

Reviewers' comments:

Reviewer's Responses to Questions

**Comments to the Author**

1. If the authors have adequately addressed your comments raised in a previous round of review and you feel that this manuscript is now acceptable for publication, you may indicate that here to bypass the “Comments to the Author” section, enter your conflict of interest statement in the “Confidential to Editor” section, and submit your "Accept" recommendation.

Reviewer #1: All comments have been addressed

Reviewer #2: All comments have been addressed

2. Is the manuscript technically sound, and do the data support the conclusions?

Reviewer #1: Yes

Reviewer #2: Yes

3. Has the statistical analysis been performed appropriately and rigorously? 

Reviewer #1: Yes

Reviewer #2: N/A

4. Have the authors made all data underlying the findings in their manuscript fully available?

Reviewer #1: Yes

Reviewer #2: Yes

5. Is the manuscript presented in an intelligible fashion and written in standard English?

Reviewer #1: Yes

Reviewer #2: Yes

6. Review Comments to the Author

Reviewer #1: Although the authors responded that they did not carry out a reproducibility analysis of the assessments and did not carry out a pilot study, I recommend that this information be included in the limitations of the study. These should even be recommendations for future studies.

As was done with the database, I suggest also making the 20 images used in the study available for public access.

Reviewer #2: Dear Authors

Congratulations on the corrections, however I still have some suggestions to improve the quality of the manuscript.

Abstract:

Please, on line 35 in the abstract, specify the number of study participants

Results

In Tables 5 and 6 identify the y-axis label (Quality of the athlete's movements). I suggest redoing the titles of Figures 5 and 6, for example, in Figure 5 it could be: Assessment of the movements of male athletes considering the color of the athlete's uniform and the sex of the evaluator. In addition, you should place information about the p value below the Figures in Note format, not in the title.

7. PLOS authors have the option to publish the peer review history of their article (what does this mean?). If published, this will include your full peer review and any attached files.

Reviewer #1: No

Reviewer #2: No

---

## [Author Response · Author response to Decision Letter 1]

25 Feb 2024

We feel great thanks for your professional review work on our article. As you are concerned, there are several problems that need to be addressed. According to your nice suggestions, we have made extensive corrections to our previous draft, the detailed corrections are listed below.

Dear expert 1:

Thank you for your valuable suggestions on this article, in response to your comments, we have made the following changes.

1.Although the authors responded that they did not carry out a reproducibility analysis of the assessments and did not carry out a pilot study, I recommend that this information be included in the limitations of the study. These should even be recommendations for future studies.

Thank you for your advice. We have added the issue of no reproducibility analysis to the limitations section, and we think that future research could explore the reliability and reproducibility of this study with experienced referees as participants.

2.As was done with the database, I suggest also making the 20 images used in the study available for public access.

Thank you for your advice. We have uploaded the figures from the experiment to a public database (figshare), specifically “10.6084/m9.figshare.25284670".

Dear expert 2:

We sincerely appreciate the valuable suggestions you have given us! In response to your suggestions, we have made the appropriate changes.

1.Abstract:Please, on line 35 in the abstract, specify the number of study participants

Thank you for your advice. On line 35 in the abstract, we added the number of study participants.

2.In Tables 5 and 6 identify the y-axis label (Quality of the athlete's movements). I suggest redoing the titles of Figures 5 and 6, for example, in Figure 5 it could be: Assessment of the movements of male athletes considering the color of the athlete's uniform and the sex of the evaluator. In addition, you should place information about the p value below the Figures in Note format, not in the title.

Thank you for your advice. We reworked the figures and made changes to the titles and notes on the figures.

---

## [Editor Report · Decision Letter 2]

7 Mar 2024

Effect of the red uniform on the judgment of position or movement used in Wushu Routine, evaluated by practitioners of the modality

PONE-D-23-24644R2

Dear Dr. Hong,

We’re pleased to inform you that your manuscript has been judged scientifically suitable for publication and will be formally accepted for publication once it meets all outstanding technical requirements.

Kind regards,

Leonardo Vidal Andreato, PhD

Academic Editor

PLOS ONE
---

## [Editor Report · Acceptance letter]

12 Mar 2024

PONE-D-23-24644R2 

PLOS ONE

Dear Dr. Hong, 

I'm pleased to inform you that your manuscript has been deemed suitable for publication in PLOS ONE. Congratulations! Your manuscript is now being handed over to our production team.

Kind regards, 

on behalf of

Dr. Leonardo Vidal Andreato 

Academic Editor

PLOS ONE